# COVID-19 preparedness and response in rural and remote areas: A scoping review

Lilian Dudley[1]*, Ian Couper[2], Niluka Wijekoon Kannangarage[3], Selvan Naidoo[2], Clara Rodriguez Ribas[3,4], Theadora Swift Koller[5]‡, Taryn Young[6]‡

**1** Division of Health Systems and Public Health, Department of Global Health, Stellenbosch University, Cape Town, South Africa, **2** Ukwanda Centre for Rural Health, Department of Global Health, Stellenbosch University, Cape Town, South Africa, **3** Health Emergencies Program, World Health Organisation, Headquarters, Geneva, Switzerland, **4** Universitat Pompeu Fabra, Barcelona, Spain, **5** Department for Gender, Equity and Human Rights, Director General's Office, World Health Organization, Headquarters, Geneva, Switzerland, **6** Division of Epidemiology and Biostatistics, Department of Global Health, Stellenbosch University, Cape Town, South Africa

☉ These authors contributed equally to this work.
‡ TSK and TY are joint last authors on this work.
* ldudley@sun.ac.za

**Data Availability Statement:** The study dataset has been published as an Excel file on the Stellenbosch University institutional research data repository, SUNScholarData. The data can be

## Abstract

This scoping review used the Arksey and O'Malley approach to explore COVID-19 preparedness and response in rural and remote areas to identify lessons to inform future health preparedness and response planning. A search of scientific and grey literature for rural COVID-19 preparedness and responses identified 5 668 articles published between 2019 and early 2022. A total of 293 articles were included, of which 160 (54.5%) were from high income countries and 106 (36.2%) from middle income countries. Studies focused mostly on the Maintenance of Essential Health Services (63; 21.5%), Surveillance, epidemiological investigation, contact tracing and adjustment of public health and social measures (60; 20.5%), Coordination and Planning (32; 10.9%); Case Management (30; 10.2%), Social Determinants of Health (29; 10%) and Risk Communication (22; 7.5%). Rural health systems were less prepared and national COVID-19 responses were often not adequately tailored to rural areas. Promising COVID-19 responses involved local leaders and communities, were collaborative and multisectoral, and engaged local cultures. Non-pharmaceutical interventions were applied less, support for access to water and sanitation at scale was weak, and more targeted approaches to the isolation of cases and quarantine of contacts were preferable to blanket lockdowns. Rural pharmacists, community health workers and agricultural extension workers assisted in overcoming shortages of health professionals. Vaccination coverage was hindered by weaker rural health systems. Digital technology enabled better coordination, communication, and access to health services, yet for some was inaccessible. Rural livelihoods and food security were affected through disruptions to local labour markets, farm produce markets and input supply chains. Important lessons include the need for rural proofing national health preparedness and response and optimizing synergies between top-down planning with localised planning and coordination. Equity-oriented rural health systems

accessed at DOI: https://doi.org/10.25413/sun.23654193.v1.

**Funding:** This research was commissioned by WHO to the University of Stellenbosch (2022/1215701-0 to IC). WHO staff had the role of instigator, co-conceptualizer and co-authors in the study. WHO drew funding for this work from grants to the Department for Gender, Equity and Human Rights, WHO/HQ, provided by the Government of Canada (7429308 to WHO;7429315 to WHO). These grants were entitled "Strengthening local and national Primary Health Care and Health Systems for the recovery and resilience of countries in the context of COVID-19" and "ACTA Health Systems and Response Connector Implementation Project: Strengthening the Foundations of COVID19 Response". The provider of these grants to WHO had no role in study design, data collection and analysis, decision to publish, or preparation of the manuscript.

**Competing interests:** The authors have declared that no competing interests exist.

strengthening and action on rural social determinants is essential to better prepare for and respond to future outbreaks.

## Introduction

Despite much global coordination and support, and calls for global solidarity, the COVID-19 pandemic exposed numerous inequities across regions, countries and populations experiencing vulnerability. The pandemic magnified chronic under-investment in health systems and in addressing health determinants in rural areas, across all countries, including those classified as low-, and middle income (LMICs) and high-income countries (HICs) [1].

Many rural populations seeking to access health services typically face challenges related to physical geography, inequity in distribution of healthcare professionals, inadequate infrastructure, a disproportionate concentration of specialised services in urban areas, lack of maintenance and shortages of equipment, undersupply of materials and drugs, and poorer communications and information technology [2–5], all of which would hamper an adequate response to COVID-19.

To support countries in their response to COVID-19, and to help address these inequities in health, the World Health Organization (WHO) 2021 COVID-19 Strategic Preparedness and Response Plan (SPRP) underlined the importance of equity-oriented approaches and meeting the needs of the most disadvantaged subpopulations, accounting for the supply and demand side barriers that influence access to effective coverage with COVID-19 related services [6].

Against the backdrop of health inequities associated with place of living, rural poverty and exclusion across countries, it was deemed important to review the available evidence about the COVID-19 health emergency preparedness and response (HEPR) in rural and remote areas. This scoping review therefore aimed to describe and explore the COVID-19 HEPR in rural and remote areas in HIC and LMIC settings. In particular we sought to assess the extent to which COVID-19 HEPR in these settings addressed the components of the WHO SPRP, and to identify the challenges, promising practices and gaps in the evidence in order to inform future policy, planning and research on HEPR in rural and remote settings. In the absence of international standards for defining rural areas, this review accepted all local or regional definitions of rural and remote areas provided in publications reviewed.

In early 2022, WHO issued a call for proposals and terms of reference for a scoping review of literature that explored COVID-19 preparedness and response in rural and remote areas. WHO's intent was to gather evidence to inform WHO's ongoing production of normative guidance and support to Member States for adequate preparedness and response to health emergencies, particularly in rural and remote areas globally. Through a competitive bidding process, a contract for the work was awarded to Stellenbosch University. Commissioned by WHO to help inform ongoing global and national processes and plans to ensure adequate preparedness and response to health emergencies, this review explores the data that is emerging for COVID-19 HEPR in rural and remote areas in all settings.

## Methods

The scoping review was guided by the Arksey and O'Malley framework for scoping reviews as modified by Levac et al. and Peters et al. [7, 8]. Scoping reviews, by definition, are a type of knowledge synthesis which follow a systematic approach to map evidence on a topic and identify main concepts, theories, sources, and knowledge gaps [9]. They do not provide a synthesis

of effects of interventions but may be used to inform the need for further systematic reviews of the literature [9]. The study protocol was published in The Open Science Framework (https://osf.io/2853a/). A glossary of terms is provided as S1 Text.

## Eligibility criteria

All scientific journal research articles, study designs, reviews, reports, book chapters, conference presentations and grey literature were included if deemed relevant to the central research question, in all UN languages and published between 2019 and the first quarter of 2022. Studies had to refer to COVID-19 preparedness and response along the categories described in the WHO SPRP 2021 [6]. In addition, studies exploring social determinants of health (SDH) and One Health in relation to COVID-19 were also included. The settings included rural and remote areas in HICs and LMICs as defined in the 2022 World Bank country classifications [10].

## Search strategy

Searches were conducted in six electronic databases in March 2022: Medline (OVID), CINAHL (EBSCOHost), Social Science Citation index (Web of Science), the Global Index Medicus (includes LILIACS), Africa-wide information (EBSCOHost), and the WHO COVID-19 Global literature on coronavirus disease. These were identified as the most relevant databases where all types of studies on COVID-19 HEPR in rural and remote areas in LMIC and HIC countries could be located. The search strategy for Medline (OVID) shown in Box 1 was used as a basis for other searches. A structured search of grey literature in targeted repositories, websites and databases was conducted in April 2022, detailed in S1 Table.

---

### Box 1. Search strategy for Medline (OVID)

Ovid MEDLINE(R) and Epub Ahead of Print, In-Process, In-Data-Review & Other Non-Indexed Citations, Daily and Versions <1946 to March 14, 2022>

1 Coronavirus/

2 coronavirus infections/ or covid-19/

3 (2019 nCoV or 2019nCoV).tw.

4 COVID.mp.

5 ("SARS coronavirus 2" or "SARS-like coronavirus" or "Severe Acute Respiratory Syndrome Coronavirus-2").mp.

6 SARS-CoV-2.mp. or SARS-CoV-2/

7 severe acute respiratory syndrome coronavirus 2.mp.

8 1 or 2 or 3 or 4 or 5 or 6 or 7

9 limit 8 to yr = "2019 -Current"

10 Rural Health/ or Hospitals, Rural/ or Rural Population/ or Rural Health Services/

11 ((disadvantage* or underserved) adj2 (area* or population* or district* or communit*)).mp.

---

12 (remote adj2 (area* or population* or district* or communit*)).tw.

13 ((Rural or pastoral or provincial) adj2 (area* or population* or district* or commu-nit*)).tw.

14 ((remote or inaccessible or isolated) adj2 (area* or population* or district* or com-munit*)).tw.

15 10 or 11 or 12 or 13 or 14

16 9 and 15

17 (preparedness or preparing or prepared).tw.

18 (readiness or willingness).tw.

19 (emergency adj2 respons*).tw.

20 (outbreak adj2 respon*).tw.

21 vaccine hesitancy.mp.

22 Vaccination/ or vaccine uptake.mp.

23 (Access* or "access to care" or knowledge or availability or provision or guidelines or "National policy" or "strategic plan").tw.

24 district nurs*.mp.

25 exp Health Services Accessibility/

26 (equity or availability).tw.

27 "Delivery of Health Care"/ or (care adj2 provision).tw.

28 health infrastructure.mp.

29 Health Personnel/ or healthcare worker*.mp.

30 Supply chain.mp.

31 (Surveillance or "epidemiological investigation").mp.

32 diagnostic services/ or clinical laboratory services/ or diagnostic screening programs/ or mass screening/ or mobile health units/

33 Case Management/

34 Health Promotion/ or Health Knowledge, Attitudes, Practice/

35 exp Contact Tracing/

36 referral.mp. or "Referral and Consultation"/

37 Intensive Care Units/ or intensive care unit*.mp.

38 "follow-up".tw.

39 Rehabilitation/

40 exp Palliative Care/ or palliative.tw

41 exp Burial/ or burial*.tw

42 exp Economics/

43 "costs and cost analysis"/ or "cost allocation"/ or cost-benefit analysis/ or "cost control"/ or "cost of illness"/

44 Economic Development/

45 ("out of pocket" or expenses or DALY or QALY).tw.

46 ((health or medical) adj2 (cost* or expense* or expenditure* or insurance)).mp.

47 Financial Stress/

48 financial hardship.mp.

49 ((social-economic or socio-economic) adj2 impact).mp.

50 "Socioeconomic Factors"/

51 exp Environment/

52 environmental impact.mp.

53 17-52/or

54 16 and 53

## Study selection

The search results were imported into Covidence [11]. Two researchers independently reviewed all the titles, abstracts and full texts in a step-wise process using the eligibility criteria described above. The article or source was included if there was a high level of agreement between the researchers. In the case of disagreement, a third researcher arbitrated. Non-English abstracts and full texts were translated to English using Google translate.

## Charting the data

An analytical framework was developed to identify standard items to be extracted. The data collection tools were piloted, and data was extracted independently by two senior researchers assisted by three research assistants, with cross checking of all data extracted. The final data was exported to Microsoft Excel Version 16 (Microsoft Corporation, USA) for storage and safety and to enable offline access [12]. The research team compared and discussed the data charting iteratively to ensure consistency. No critical appraisal was undertaken to assess the quality of the included studies in keeping with the purpose and approach of a scoping review.

## Collating, summarising and reporting the results

A descriptive analysis of the data provided an understanding of the distribution of the categorical variables and main characteristics of the dataset, and is reported in the results section as frequencies and percentages of studies by World Bank category, country setting, year of publication, study population, particular risk groups, study design, and WHO SPRP pillars as listed in Box 2 below.

Box 2. WHO SPRP pillars

1. Coordination, planning, financing and monitoring;

2. Risk communication, community engagement and infodemic management;

3. Surveillance, epidemiological investigation, contact tracing and adjustment of public health and social measures;

4. Points of entry, international travel and transport, and mass gatherings;

5. Laboratories and diagnostics;

6. Infection prevention and control, and protection of the health workforce;

7. Case management, clinical operations, and therapeutics;

8. Operational support and logistics, and supply chains;

9. Maintaining essential health services and systems; and

10. Vaccination.

Synthesis of the content was guided by the study aims and objectives, and each pillar was analysed in terms of key findings, successes and enablers, challenges and barriers to COVID-19 preparedness and response, gaps in the evidence and further research required. A thematic analysis was conducted and is presented along the following main deductive themes that reflect the final selected SPRP pillars as well as Social Determinants of Health: (i) Coordination, planning, financing and monitoring; (ii) Risk communication, community engagement (RCCE) and infodemic management; (iii) Surveillance, epidemiological investigation, contact tracing and adjustment of public health and social measures (PHSM); (iv) Laboratories and diagnostics; (v) Infection prevention and control (IPC), and protection of the health workforce; (vi) Case management, clinical operations, and therapeutics; (vii) Maintaining essential health services and systems (MEHSS); (viii) Vaccination; (IX) Social Determinants of Health (SDH). Within each of these themes we used an inductive approach drawn from grounded theory to analyse the data [13]. The data within each theme was coded using both hand-coding and nVivo software version 12 to identify emergent subthemes, which were discussed iteratively within the study team. We described the subthemes of each of the main themes in the narrative, illustrating this with study examples.

The Preferred Reporting Items for Systematic Reviews and Meta-Analyses -Extension for Scoping Reviews (PRISMA-ScR) guidelines shaped the reporting [14]. The completed PRISMA-ScR Checklist is provided as S1 Checklist.

## Results

The search identified a total of 5 668 records, of which 293 were included in the review after the sequential removal of duplicates and screening of the titles, abstracts and full texts. See Fig 1. The main reason for excluding records were that they were not rural or remote (130 publications), were opinion articles, commentaries, letters, interviews or editorial reviews (113 publications), or did not include COVID-19 preparedness or response (72 publications). A list of the included studies is provided as S2 Table.

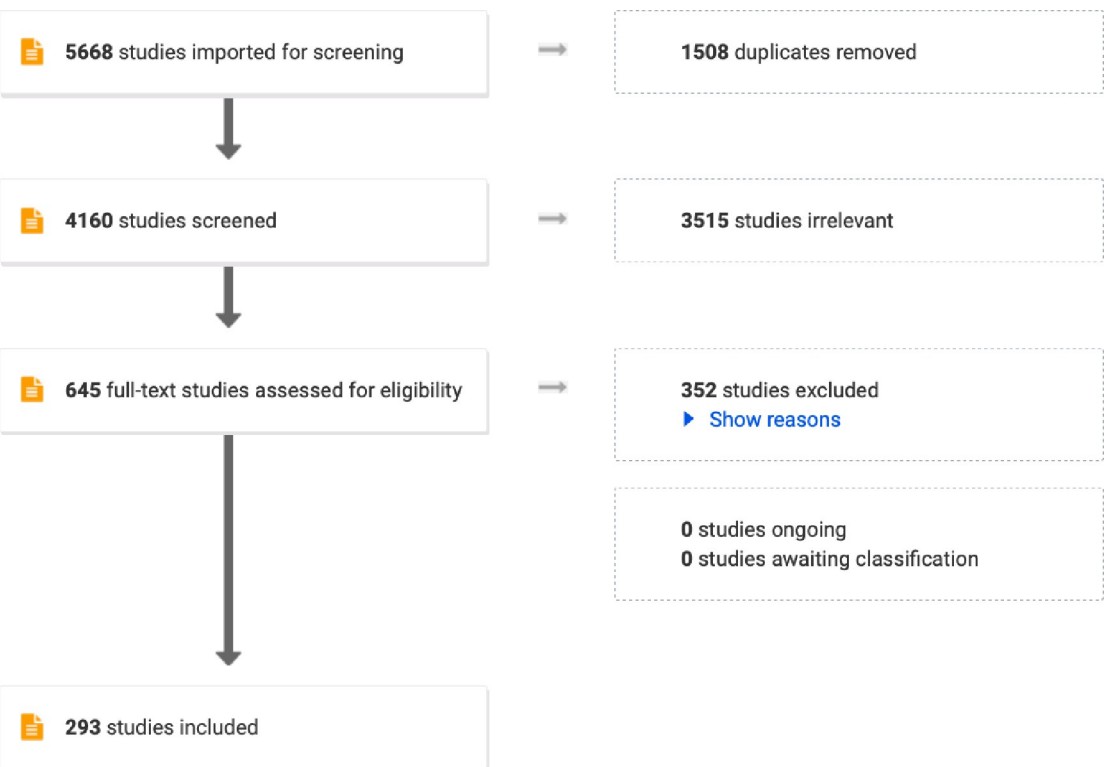

**Fig 1. PRISMA flow diagram of the screening and selection of studies.**

### Description of included studies

Of the 293 included studies, 49 (16.7%) were published in 2020, 189 (64.5%) in 2021, and 55 (18.8%) in the first 3 months of 2022. The majority were conducted in HICs (54.5%) and MICs (36.2%) (Fig 2). They focused entirely on rural areas (157, 53.6%) or included a rural sub-analysis within a national or regional study 136 (46.4%). Of the WHO regions, almost half (135, 46.1%) were from the Americas, 48 (16.4%) from the Western Pacific, 35 (12%) from South East Asia, 35 (12%) from Africa, 21 (7.2%) from Europe, and 11 (3.8%) from the Eastern Mediterranean; 10 (3.4%) included countries across more than one WHO region. The five countries with the most included studies were the USA (105, 35.8%), India (25, 8.5%), China (20, 6.8%), Australia (16, 5.5%) and Canada (10, 3.4%).

Of the 10 WHO SPRP pillars, many publications were on Maintaining essential health services and systems (MEHSS) (63, 21.5%) and Surveillance, epidemiological investigation, contact tracing and adjustment of public health and social measures (PHSM) (60, 20,5%) (Fig 3). We found very few studies on Points of entry, international travel and transport, and mass gatherings (3 publications), Operational support and logistics, and supply chains (3), and One Health (2) and excluded these in further analysis.

The proportion of HIC and LMIC studies in each included pillar varied, with HIC studies forming the bulk of the MEHSS, Vaccination, Case management and Laboratory and diagnostics groups. There was a more even distribution between HICs and LMIC's in the Surveillance and PHSM and Infection prevention and control (IPC) groups, and more LMIC publications on Risk communication and community engagement (RCCE) and SDH Table 1.

The main study designs were cross sectional (91, 31.1%), case reports or case series (56, 19,1%), qualitative methods (46, 15.7%), mixed methods (25, 8.5%), data modelling including

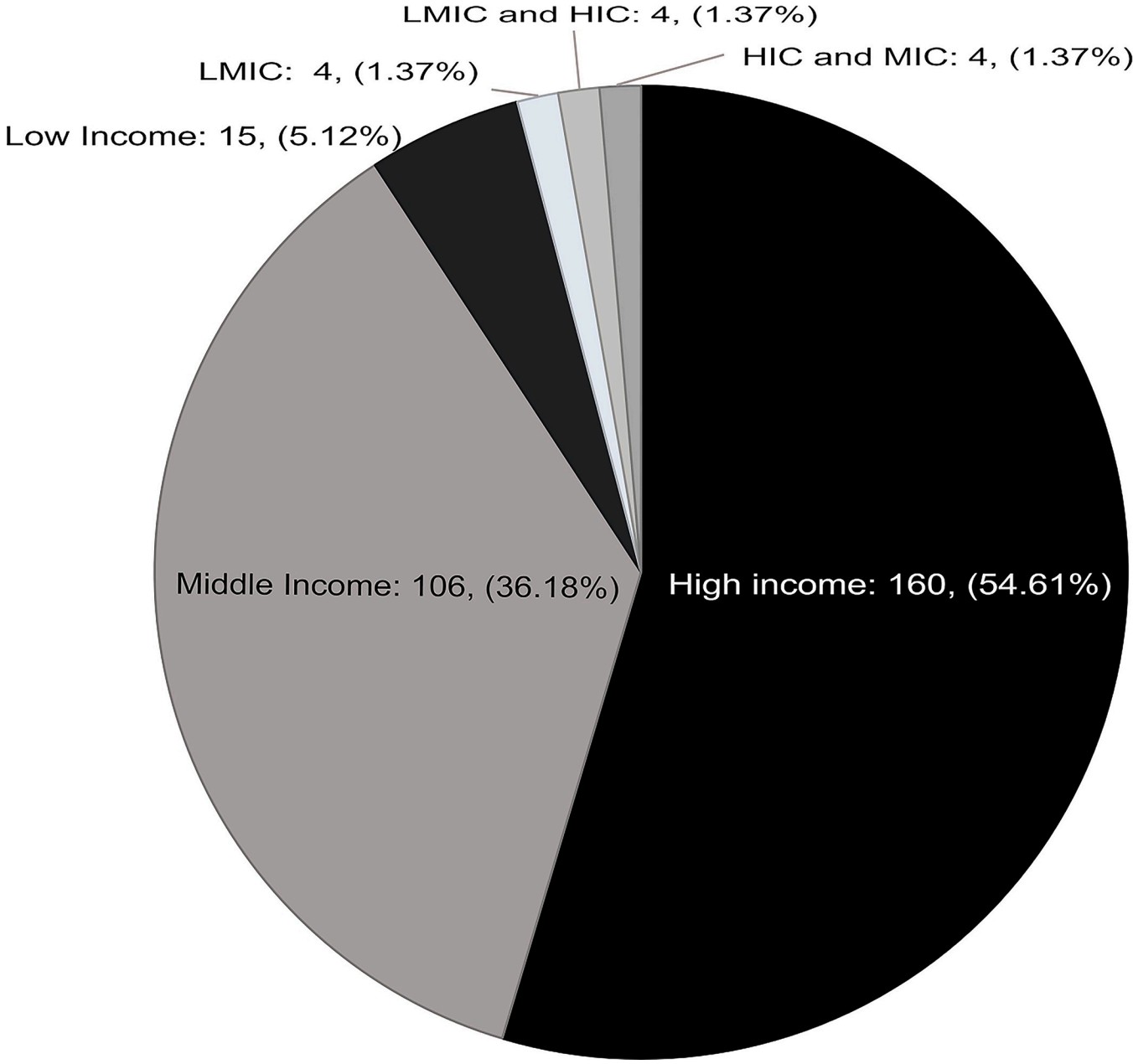

**Fig 2. Distribution of included studies by World Bank country income level categories.**

geospatial modelling (22, 7.5%), and cohort studies (15, 5.1%). Very few randomised control trials (4, 1.4%), non-randomised experimental studies (8, 2.7%), systematic reviews (3, 1.0%), or other types of reviews (10, 3.4%) were found.

The study populations were mainly a community (126, 42%), patients or healthcare users (92, 31.4%), or healthcare workers (41, 14.0%). Few studies focussed primarily on policy-makers (10, 3.4%), combinations of different populations (13, 4.4%), informal health workers (3, 1.0%) or populations outside of the health sector such as teachers, police, farmworkers and other essential workers (8, 2.7%). Many studies did not identify a specific risk group (155, 53.5%). Of the 138 studies that did, health care workers (42, 30.4%), indigenous communities

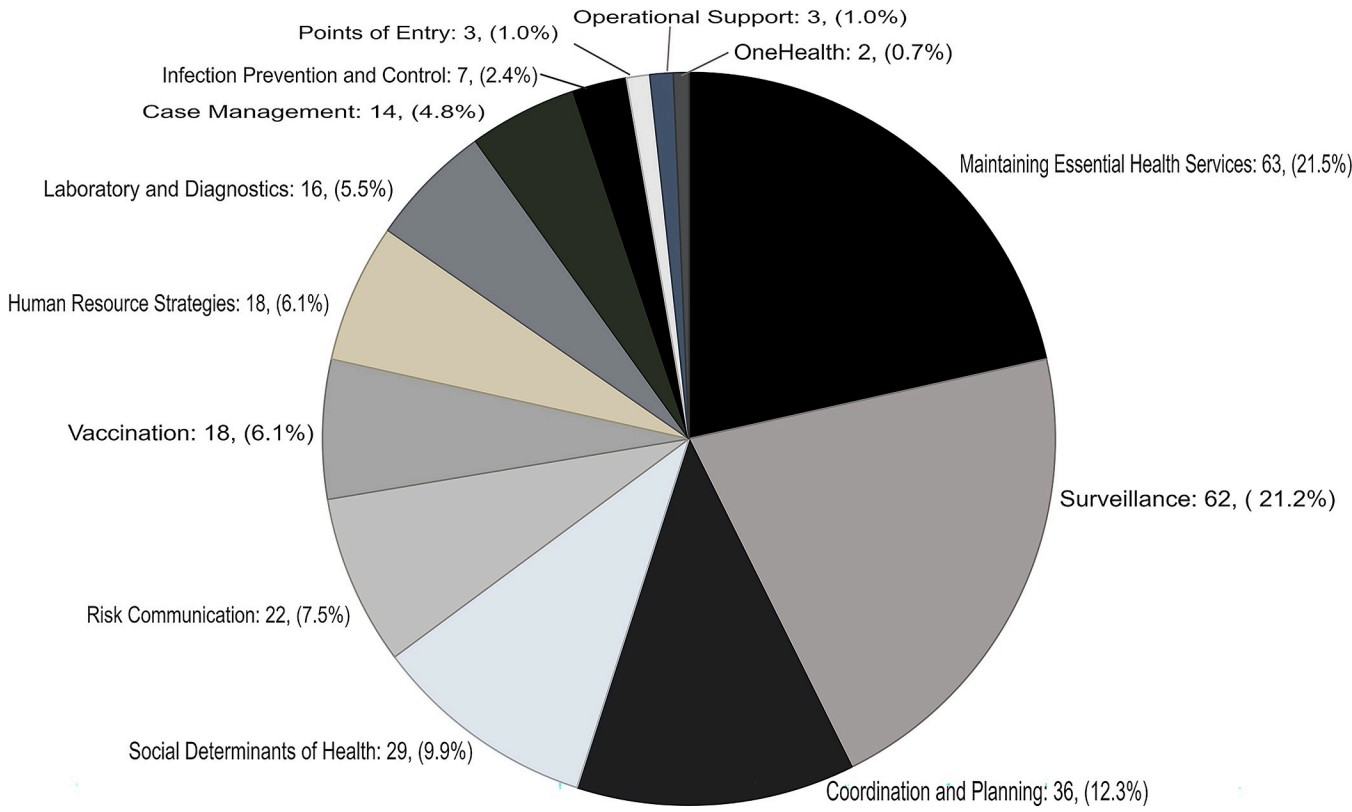

**Fig 3. Distribution of included studies across themes.**

or remote islanders (23, 16.7%), chronic disease patients (20, 14.5%), and older people (16, 11.6%) were the main risk groups. Farmers and farmworkers (6, 4.3%), other essential workers (7, 5.0%), women (6, 4.3%), children and adolescents (5, 3.6%), ethnic groups (4, 3.0%) and migrants or refugees (2, 1.4%) were specified in a few studies.

## Thematic analysis

This section reports on the thematic analysis of the included studies. We combined a deductive and inductive approach, using the SPRP pillars and SDH as the main themes, and identifying

**Table 1. Distribution of studies across themes and World Bank country groups.**

| SPRP Category (Themes) | High income countries (HIC) | | Middle income countries (MIC) | | Low income countries (LIC) | | Mixed (HIC, MIC, LIC) | | Total (n) |
|---|---|---|---|---|---|---|---|---|---|
| | n | % | n | % | n | % | n | % | |
| Maintenance of essential health services & systems (MEHSS) | 50 | 79,4 | 11 | 17,5 | 1 | 1,6 | 1 | 1,6 | 63 |
| Surveillance, epi investigation & public health and social measures | 26 | 41,3 | 25 | 39,7 | 6 | 9,5 | 3 | 4,8 | 60 |
| Coordination and planning | 18 | 56,3 | 9 | 28,2 | 3 | 9,3 | 2 | 6,0 | 32 |
| Case management, clinical operations and therapeutics | 19 | 63,3 | 10 | 33,3 | 0 | 0,0 | 1 | 3,3 | 30 |
| Social determinants of health (SDH) | 6 | 20,7 | 17 | 58,6 | 2 | 6,8 | 4 | 13,8 | 29 |
| Risk communication, community engagement & infodemic management | 8 | 36,4 | 13 | 59,1 | 1 | 4,5 | 0 | 0,0 | 22 |
| Vaccination | 13 | 72,2 | 5 | 27,8 | 0 | 0,0 | 0 | 0,0 | 18 |
| Laboratory and diagnostics | 9 | 56,3 | 7 | 43,7 | 0 | 0,0 | 0 | 0,0 | 16 |
| Infection prevention and control (IPC) & protection of health workers | 8 | 50,0 | 5 | 31,3 | 2 | 12,5 | 1 | 6,3 | 16 |

subthemes which emerged from the data within each of these main themes. See Table 2. In describing the subthemes we provide examples of studies which illustrate the subthemes. These referenced studies do not necessarily provide evidence of effect nor are they representative of the entire COVID-19 response in particular countries or regions mentioned.

**Coordination, planning, financing and monitoring.** The 32 studies in this SPRP pillar dealing with overarching governance issues, described coordination, planning, financing and monitoring at various levels from communities, to local, state, national and international levels. The main subthemes which emerged were:- i) Inclusion of rural and remote areas in policies; ii) Coordination between national policies and rural responses; iii) Operational readiness in rural areas, iv) Factors which enhanced local coordination of preparedness and response; and v) Financing.

*Inclusion of rural and remote areas in policies.* The review identified few policies and guidelines which specifically included an approach to rural or remote areas, or provided information on COVID-19 preparedness and response in rural populations. At the level of regional guidance to support national COVID-19 responses, the PAHO guidance on strengthening the first level of care had an explicit embedded focus on rural and remote areas to promote targeted planning at that level [15]. In HICs, the USA National COVID-19 Preparedness Plan of 2022, [16] was an example of a policy which provided detailed guidance for rural and remote areas, and identified rural challenges. In some LMIC countries with large rural populations, national policies identified through the review appeared not to differentiate between rural and urban areas [17, 18]. Examples of challenges were described in reviews of India's national policies in which authors identified overall health policy and system inadequacies that could hinder an effective COVID-19 response, but also highlighted the capacity for successful responses in states with large rural populations such as Uttar Pradesh and Rajasthan [19, 20].

*Coordination between national policies and rural responses.* A few studies reported tensions between national policy and local responses where authors reported that top down, hierarchical national responses did not provide adequate space for local innovation and adaption [21, 22]. In some settings this was resolved to create a balance between top down and bottom-up responses, but in others, a rigid national approach and extreme border controls impacted negatively on rural and remote communities. Local autonomy was also impeded by structural factors such as workforce turnover, limited funding, medicolegal concerns, and a lack of access to information.

*Operational readiness in rural areas.* In the USA, operational readiness assessments at various levels found that rural areas had fewer health professionals, hospitals and intensive care units (ICU), despite having more vulnerable populations and reporting more deaths per population than urban areas [23–25]. Rural health services including Emergency Medical Services (EMS), community pharmacists and home health agencies were less prepared, and facilities were predicted to reach maximum capacity faster than in urban areas [26, 27].

*Factors which enhanced local coordination of preparedness and response.* Despite the tensions and local constraints, studies suggested that a balance between top down and bottom up responses can be achieved by building on well-established social structures, and a strong sense of social solidarity [21, 28]. Good relationships between government services and local leaders were critical in influencing attitudes and behaviours of rural populations [21, 28]. Planning and implementation by local multisectoral committees also ensured appropriateness of strategies, and collaborative leadership enhanced collective action in community governance and effective responses [29].

Rural COVID-19 responses relied heavily on local level institutions whose effectiveness and efficiency depended on the nature of those institutions, and their capacity, legitimacy and a history of policy interventions [30]. Important organisational attributes that enabled

**Table 2. Thematic and subtheme categories and a summary of key findings.**

| | Main Theme (n = study numbers) | Subthemes | Summary of Key Findings |
|---|---|---|---|
| i. | Coordination, planning, financing and monitoring. (32) | Inclusion of rural areas in policies | Limited evidence of adaptions for rural and remote settings in policies and plans. There were a few examples of good practice at regional and national levels in HIC and LMIC settings. |
| | | Coordination between national policy and local responses | Tensions between levels due to 'top down' approaches. This undermined local autonomy, but was resolved in some settings to create a balance between top down and bottom up. |
| | | Operational readiness in rural areas | Rural health systems and services were less equipped and prepared, lacking appropriate staffing, infrastructure, and supplies. |
| | | Factors affecting the coordination and planning of local responses | Local responses were affected by structural conditions, particular leadership and organisational attributes, and trust and collaboration between government and communities. |
| | | Financing | Health services and households in some settings were adversely affected by inadequate financial protection |
| ii. | Risk Communication, Community Engagement (RCCE) and infodemic management. (22) | Access to risk communication | Rural communities received less risk communication, received it later, and in forms that were less appropriate for their context. |
| | | Differing needs and contexts for risk communication | An understanding of the extent of rurality or remoteness, poverty, educational levels, cultural and linguistic realities and experiences of social exclusion is necessary to inform appropriate risk communication. |
| | | Engagement and roles of local leaders and organisations | Local leaders, organisations and health services are key to engaging local communities in HEPR. However, national RCCE campaigns bypassed them reducing trust in risk communication. |
| | | Effects of risk communication and community engagement | The impact of RCCE on behaviour was moderated by belief systems, cultural and social norms, politics, the constraints of living conditions and access to health care in rural settings. |
| iii. | Surveillance, epidemiological investigation, contact tracing and adjustment of public health and social measures (PHSM). (60) | Surveillance systems, outbreak investigation and control measures | Innovative surveillance included integrated electronic health information systems and community based surveillance. Contact tracing and quarantining **all** contacts worked, and along with shielding of vulnerable groups was more socially acceptable than blanket lockdowns in rural areas. |
| | | Prevention strategies | Non pharmaceutical interventions (NPI's) were applied less in some rural communities but worked better with the involvement of local promoters providing in-person reinforcement. |
| | | Social measures | State social welfare and community social support initiatives buffered the negative impact of COVID-19 control measures and enhanced compliance with lockdown measures. |
| iv. | Laboratories and diagnostics. (16) | Access to laboratory testing | Rural communities had less access to testing, with several barriers reducing access. Non -traditional medical and non–medical testing sites, mobile testing and eHealth technology improved rural access |
| | | Types of tests and samples | Rapid molecular tests and point of care testing decreased turnaround times in rural settings. Pooled sample RCT testing, and rapid tests of waste water and sewage was used for early detection in some locations. |
| | | Radiology | One study illustrated how radiological images from a repository of rural COVID-19 patients assisted decision making on clinical diagnosis and management. |
| v. | Infection Prevention and Control (IPC) and protection of the health workforce. (16) | Water, sanitation and hygiene (WASH) | Support for provision of basic WASH at scale was weak for rural and remote communities. WASH was inadequate in COVID-19 treatment facilities in some LMIC rural settings. |
| | | IPC attitudes and practices | IPC was constrained by a lack of PPE, high workloads, and inadequate preparation of health providers. |
| | | Protection of health workers | Health workers feared infection; worked while waiting for results; and some lacked paid sick leave protection. |

(*Continued*)

**Table 2.** (Continued)

|  | Main Theme (n = study numbers) | Subthemes | Summary of Key Findings |
|---|---|---|---|
| vi. | Case management, clinical operations, and therapeutics. (13) | Rural hospital services | Rural hospitals, largely led by generalists, had insufficient support from higher levels, but facilitated effective local responses and models of care to ensure linkages across hospital levels. Equity was an ongoing concern. |
|  |  | Rural PHC services | Access to primary care was constrained. Rural PC health services and practitioners reorganised practices to triage patients; managed patients remotely; took on outbreak control functions; and collaborated more across disciplines. Pharmacists and CHW's filled gaps in care, assisting with HEPR in communities. |
|  |  | Preparation and practices of health workers | Practitioners reported a lack of preparedness causing fear and anxiety. They collaborated more in teams, improvised, and built trust with communities. Innovative training programs using digital technology supported their preparation. New models of care delivery improved access to testing, care and support. |
|  |  | Therapeutics | Limited to two studies of monoclonal antibody use in rural USA, and one of herbal medicine use in Vietnam. |
| vii. | Maintenance of Essential Health Services and Systems (MEHSS). (63) | Telemedicine/Telehealth | Telemedicine uptake increased, but was lower than in urban settings with many barriers in rural settings, more so in LMIC's. Where it was feasible, it improved access to a wide range of COVID-19 prevention, treatment and rehabilitation services, with many benefits for patients and providers. |
|  |  | Access to medicines and other health care | Sustained access to medicine was enabled by providing longer periods for 'take home' medication; home deliveries; community 'pick up' points; and the use of GIS to plan medication distribution in rural areas. Other adaptations included expansion of mental health care by building capacity in communities and for health professionals; and reorganising mobile EMS services in remote areas. |
| viii. | Vaccination (18) | Vaccine coverage and effects | Vaccine coverage was lower in rural than urban areas. Shortages of health professionals and infrastructure contributed was a key factor. Vaccine uptake in rural health care workers was also lower in several settings. |
|  |  | Vaccine hesitancy | Key concerns of rural communities about vaccines included the novelty of the vaccine, side effects, the number of doses and the role of industry. Accessibility and availability was a concern in LMIC's. Information directly from health professionals or health authorities increased confidence in the vaccines. |
|  |  | Strategies to improve vaccine acceptance and uptake | These included information and communication strategies, integration of COVID-19 vaccination in routine medical care, and alternate rural delivery sites including to homes, workplaces and schools. |
| ix. | Social Determinants of Health (SDH). (29) | Impact of COVID-19 on SDH in rural areas | COVID-19 and control responses contributed to greater deprivation, loss of income, disruptions to food systems and food insecurity, and reduced access to health care and other services in rural areas. Women, small farmers, and larger low income families were worst affected. |
|  |  | Food security responses | Food insecurity was widespread in rural settings. State strategies to prioritise agriculture and food security, including incentives to increase production and state supported food assistance, reduced levels of food insecurity. Community responses included food banks, food parcels, food gardening, and food sharing. Household coping strategies included reduced food consumption and dietary changes. |
|  |  | Social and economic responses | Social and economic support included cash transfers, emergency supplies, social prescribing, social solidarity through reciprocal practices and technological innovations. Cash transfers appeared to benefit urban residents more. Several communities drew on traditional practises to strengthen community resilience. |
|  |  | Role of digital technology | Digital technology contributed to increased self-efficacy and well-being in rural areas through access to learn new skills, social media to ask for or provide help, enabled bartering in digital form to sell produce, communicate and share food and connect with markets. |

appropriate COVID-19 responses included strong leadership, effective communication, and multi-disciplinary collaboration, as well as operational elements such as a well-functioning incident command system with clear roles and responsibilities, and regular communication with health service employees. Good responses were based on well-defined and well implemented plans, included expanding intensive care capacity; provision of COVID-19 information, screening, testing and follow up services for geographically dispersed populations; coordination across large diverse organisations; and the integration of community-based responses. A strong degree of trust and collaboration between local government and communities facilitated more appropriate responses to the pandemic [30].

Studies in indigenous communities found that structural conditions limited local adaptive capacity to COVID-19, with high dependence of communities on public transfer payments as illustrated in a study on Torres Strait Islanders [31]. Lessons learnt were to ensure the participation of indigenous peoples as partners in decision-making regarding the planning, development, and implementation of programmes and in the development of preventive measures against COVID-19 [31].

*Financing.* Of the limited studies addressing health financing, and within that, financial protection, an important finding of a study in China was that health insurance did not adequately cover the costs of COVID-19 hospital care for rural households, who were at risk of catastrophic expenditures [32]. Health services in rural areas in USA were also affected financially through loss of revenue [16, 27].

**Risk communication, community engagement and infodemic management.**   Twenty-two publications described a range of communication platforms including traditional media such TV, radio, print media, leaflets, loudhailers; digital media such as SMSs, webinars, social media, and videos; interpersonal communication; or a mix of different approaches and platforms. Subthemes included i) Access to risk communication; ii) Differing needs and contexts for risk communication; iii) Roles of local leaders and organisations; and iv) Effects of risk communication.

*Access to risk communication.* Rural and remote communities received less risk communication, received it later, and received it in forms that were less appropriate to their context, needs and capacity to access and apply the information, as illustrated in studies from China and Australia [33, 34].

*Differing needs and contexts for risk communication.* The articles suggested that risk communication needs and contexts differ across target groups in HIC and LMIC countries, and that an understanding of the extent of rurality or remoteness, poverty and education levels, and experiences of social exclusion is required to appropriately inform interventions. Within rural areas, there were subpopulations who had different cultural and linguistic realities and for whom messaging required specific adaptations [35]. A study from Nigeria reported that rural communities had less access to digital media resources and relied more on traditional communication tools such as radio, TV, outreach visits and interpersonal communication [36].

*Roles of local leaders and organisations.* Several studies suggested that the role of local leaders, organisations and health services was key to engaging rural communities but was neglected in national 'campaigns', reducing the trust in risk communication [34, 37, 38]. In rural USA local Community Health Worker (CHW) organisations used advocacy to bring the issues of diverse communities to the forefront during planning on how to address the pandemic [39].

*Effects of risk communication.* Select studies reported that risk communication improved awareness, knowledge and some attitudes. [40, 41], and that the use of multiple platforms and channels of communication worked better than single platforms [42–44]. The impact on behavior was moderated by belief systems, cultural and social norms, politics, and the

constraints of poverty, living conditions, and limited access to health care in several rural and remote communities [45–47].

**Surveillance, epidemiological investigation, contact tracing and adjustment of public health and social measures.** With 60 publications providing evidence on this theme, the analysis is presented along three sub-themes: i) COVID-19 surveillance systems, outbreak investigation and control measures, including contact tracing, quarantine, isolation and lockdowns; ii) Prevention strategies particularly Non-Pharmaceutical Interventions (NPIs) such as mask wearing, social distancing and handwashing, and iii) Social interventions, or assessed methods of assessing risk and protecting particular risk groups.

Religious festivals, movement of migrant workers, returning younger people from urban areas, and tourism were identified as factors in the spread of COVID-19 to rural areas [20, 48–50]. Whilst in some cases less severe disease was reported in rural patients, in others excess death measured as Infection fatality ratio (IFR) was higher in rural communities and lower in urban communities [51, 52]. The risk of hospitalisation was higher in rural areas of certain HICs, but hospitalisation rates were lower in rural areas in some LMICs [52–54].

*COVID-19 surveillance systems, outbreak investigation and control measures, including contact tracing, quarantine, isolation and lockdowns.* To contribute to meeting the need for reliable data on COVID-19 prevalence, electronic surveillance platforms were established, drawing on existing laboratory and patient data systems as seen in the case of Scotland, United Kingdom [55]. Community based surveillance studies were conducted in HIC and LMIC rural settings to improve estimates of COVID-19 seroprevalence [56, 57]. Two studies of community surveillance in rural areas of LMICs using rapid tests found a much higher prevalence of COVID-19 infection than reported through routine laboratory-based surveillance [52, 58]. CHWs assisted with surveillance, screening and detection of potential cases in the community or in returnees to villages and referring them for testing. They also assisted in identifying high risk individuals in the community, in contact tracing, supporting quarantine measures, and educating the community about NPI [54, 59–61].

Outbreak control in indigenous nations in the USA demonstrated the importance of cross-trained personnel who integrated tasks along the testing–tracing continuum, enabling the rapid identification of new cases and contacts. They also demonstrated collaborative efforts of Tribal government, federal partners and community members to respond to the pandemic by implementing surveillance, response coordination and communication systems [62].

Outbreak investigations in high-risk food processing occupational settings in rural USA found that screening, testing and quarantine measures of incoming workers was inadequate, and clusters of outbreaks occurred before control measures could be implemented [63, 64]. Outbreak investigations in rural health and tourism settings faced unique challenges, as reported by studies in the US and Australia, with long delays in obtaining test results limiting case identification and contact tracing efforts, and placing strain on the limited staff resources, resulting in the closure of health facilities [65–68].

A systematic review (47 studies across all settings) of contact tracing for communicable diseases, including COVID-19 studies and rural areas, found that contact tracing was an effective public health tool across a range of diseases, settings and approaches [69]. Data modelling of contact tracing and quarantine found that increasing the fraction of traced contacts decreased the size of the epidemic, and that quarantining all the traced contacts was more effective than quarantining only test positive traced contacts [70, 71]. Although contact tracing apps held promise, lower adoption rates reported in rural USA limited their utility [72].

Diligent contact tracing, strict isolation of cases and quarantining of all contacts, as well as shielding of vulnerable populations such as older people, reduced community transmission in rural areas of the UK and China, and were more acceptable than the blanket lockdowns

implemented in urban areas [73, 74]. Although 'shielding of older people' was a challenging concept in rural communities in Sudan, it was acceptable if applied within the household rather than separating older family members geographically [75]. Studies from Sierra Leone and China showed that the negative psychological effects of lockdowns impacted older rural residents more than younger people in these areas, particularly those who were dependent on adult children working in urban areas for social and financial support as well as females in rural areas [76, 77].

*Prevention strategies particularly Non-Pharmaceutical Interventions (NPIs) such as mask wearing, social distancing and handwashing.* Non-pharmaceutical interventions (NPIs), such as physical distancing and mask wearing, were effective in reducing symptomatic SARS-CoV-2 infections in a trial in rural Bangladesh and a mathematical modelling study in Ethiopia [78, 79]. In-person reinforcement by 'mask promoters' was important to adopting NPI's in the Bangladesh trial [78]. Modelling studies found that NPIs were more effective than strict national lockdowns or other strategies in both rural and urban areas [80, 81]. However, several studies reported that rural residents tended to apply NPI preventive measures less than urban residents which was attributed to lower education levels, poorer access to COVID-19 information, different perceptions and cultural norms, the inability to 'work from home' on farms and in rural areas, and the need in remote communities to obtain supplies [52, 61, 82, 83]. In contrast, a study in Brazil found that urban residents practised less social distancing than rural residents [84]. In Iran, public health officials, CHWs, agricultural extension workers, agricultural cooperatives and unions were reported to play an important role in rural settings in advising households on COVID-19 prevention and control, in setting up community-based networks and building local capacity [85].

*Social interventions, or assessed methods of assessing risk and protecting particular risk groups.* Social interventions were also shown to be important in reducing the negative side-effects of prevention and control measures in rural areas. For example, in South Africa, one study described how government-supported social welfare programmes aimed to buffer interruptions in income and healthcare access during local outbreaks [86]. Ensuring access to food supplies (e.g., WhatsApp groups with local grocers, free food for poorer households) and financial incentives were reported to enhance compliance with lockdown measures in Chinese villages [87]. Community initiatives included women in rural and urban communities with social and financial capital offering loans to other women, who needed assistance to feed their families. One study also indicated that community cohesion was positively associated with better mental health under lockdown [77].

Increased risks of violence against women and children have been documented as a negative side effect of prevention and control measures. A Canadian study documented the effects of COVID-19 on Inter-Personal Violence (IPV) in remote and rural communities, including indigenous communities. Reduced access to local care and limited access to appropriate tele-health services resulted in women who were seeking care being sent away from their communities [88].

**Laboratories and diagnostics.** Sixteen publications described subthemes of i) Access to laboratory testing, ii) Types of tests and samples, and iii) Radiology in COVID-19 screening, testing and care.

*Access to laboratory testing.* Rural communities were reported to have less access to testing than urban communities [89, 90]. Barriers to testing included uncertainty regarding testing guidelines, where to go for tests, accessible testing locations, perceptions that testing was painful, and long waiting times for results [91]. Response strategies included expanding testing to non-traditional medical and non-medical sites, mobile testing and integrating eHealth technology for testing. In Lebanon, for example, portable primary health clinics provided COVID-

19 testing, and linked the patients via telemedicine to doctors as required [92]. In Canada, rural, remote, urban and First Nations communities reported a high level of acceptability and satisfaction with a Virtual Triage and Assessment Centre established to increase access to testing and care [93].

*Types of tests and samples.* The availability of rapid molecular tests and the implementation of rapid point of care testing dramatically decreased the turn-around time for results in rural communities as illustrated in studies in the USA [94, 95]. Innovations in specimen collection media promised to improve the feasibility of molecular testing in remote or rural communities by reducing constraints experienced in the transporting of specimens [96].

One study from India suggested that pooled sample RT PCR testing was a reliable, effective approach for large scale screening of samples in low prevalence rural communities [97]. Studies from Brazil and Canada explored the use of rapid tests of waste water or raw sewage as early detection methods for COVID-19 in rural communities with a low prevalence of SARS CoV-2 [98, 99].

*Radiology in COVID-19 screening, testing and care.* A repository of radiological images from rural COVID-19 patients in the USA provided information on typical radiographic changes to assist in the diagnosis of COVID-19 and clinical decision making [100].

**Infection prevention and control, and protection of the health workforce.** Sixteen publications described i) Water, sanitation and hygiene (WASH), ii) Attitudes and practices in IPC, and iii) Protection of health workers.

*Water, sanitation and hygiene (WASH).* A review of COVID-19 WASH measures across 84 countries found wide variations, with basic hygiene promotion and IPC widely adopted, but support to promote basic access to WASH services at scale was weak, particularly for vulnerable households in rural areas and small towns [101]. In Ethiopia, a study reported deficiencies in the provision of WASH in 35 temporary COVID-19 isolation and treatment centres, particularly in rural areas, with daily water supply interruptions common, few functional laundries, insufficient hand washing stations, and a general shortage of Personal Protective Equipment (PPE) [102].

*Attitudes and practices in IPC.* In several countries, rural IPC and other health workers reported major challenges including a lack of access to PPE, overwhelming workloads and multiple responsibilities, which impeded the implementation of IPC measures and contributed to fear of infection for themselves and their patients [103–107].

IPC strategies in some rural areas in the USA included separate temporary quarantine facilities for COVID-19 screening of patients seeking entry into a residential addiction treatment program to prevent outbreaks in the centre; and temporary structures in rural areas for large-scale decontamination and re-use of PPE for hospitals and clinics to mitigate the risk of PPE procurement issues [108].

A hospital infectious diseases unit in rural Uganda was rapidly organised with multidisciplinary teams and access to speciality care to admit COVID-19 patients. However, as numbers of COVID-19 patients increased, patients had to be admitted to other wards, and COVID-19 care had to be integrated with cross-trained staff requiring capacitation in IPC [109].

An analysis of the first nosocomial outbreak of COVID-19 within a healthcare environment in Australia found that rural healthcare environments and staff were ill-equipped for the pandemic. It also suggested that health professional presenteeism, with employees continuing to work while awaiting COVID-19 test results because of staff shortages, contributed to the COVID-19 outbreak and affected health service provisions [66].

*Protection of health workers.* A review of Workman's Compensation in the USA found that although paid sick leave reduced new COVID-19 cases and COVID-19 activity in health

workers and other essential workers, rural agricultural and food processing workers had inadequate sick leave protection contributing to housing and food insecurity [110].

**Case management, clinical operations, and therapeutics.** Thirty publications described i) Rural hospital and primary care services for COVID-19, ii) Preparation and practices of rural health workers, and iii) Delivery of particular therapeutic agents in these settings.

*Rural hospital and primary care services for COVID-19.* Rural hospitals, in many cases led by generalists, provided essential health care for mild and moderate cases of COVID-19, taking pressure off higher-level facilities [111–114]. They faced challenges in operating at the margins at a time when communication channels were disrupted, and support from higher levels was variable, often with limited understanding of the rural hospital context [112]. An Australian study found that approaches which worked in urban areas such as designated COVID-19 hospitals, were less appropriate in rural areas where distance and travel were constraints [115]. Local hospital leadership in South Africa, New Zealand and the USA facilitated effective local responses, and models were developed such as a 'hub-and-spoke system' in which sicker patients were routed to a central rural or urban hospital [111, 112, 114]. Specific aspects of rural hospital preparedness included strengthening IPC, scaling down existing services such as elective surgery, workforce management, rapid expansion of telehealth, increased use of telecommunication with daily system wide briefings, the importance of training and exercises, feedback systems, and surge planning. Despite the responses at rural hospitals, equity concerns persisted about access to hospital care, particularly advanced care in rural areas [112, 114].

Primary care responses to COVID-19 varied, with rural Community Health Centres (CHCs) in the USA providing significantly fewer COVID-19-related care visits than urban CHCs, raising concerns about access to care [116]. A South African study also identified concerns about access to primary care for COVID-19 patients in rural areas [114]. Rural primary care physicians in Germany re-organised their practices to screen and triage patients, as well as managing patients remotely by telephone or video. They also observed a higher probability for more severe disease in COVID-19 patients in rural areas [117, 118]. In China, one study reported that primary care practitioners took on key public health roles of tracing, screening, and educating in rural areas [119]. According to the study, their role in treating patients was reduced as they did not transition to remote consultations and many patients were diverted to specialist COVID-19 clinics [119]. In Egypt tele-pharmacy teams, collaborating with prescribers in rural areas, were found to improve access to safe therapeutics, health education, ongoing monitoring and early detection of adverse events or deterioration in COVID-19 patients [120].

Multiple studies highlighted the role of community health workers (CHWs). Nurse-led Family Health Units in Brazil adopted strategies for prevention, health promotion, dealing with suspected COVID-19 cases and maintaining attendance of priority programs, while community health agents visiting homes played an important role in the response [121]. In the USA, CHWs also filled gaps in the provision of care to rural communities, using innovative ways to reach their clients, including video meetings, phone calls, and face-to-face meetings adhering to social distancing guidelines [39]. In India, the participation of CHWs helped ease the strain on frontline medical workers, augmented the capacity of the public health-care infrastructure and brought services closer to the community [122]. One study also raised the importance of adequately capacitating CHWs for the task, as otherwise their engagement could be counterproductive: the study suggested that some CHWs in Nigeria had a poor level of knowledge and practice of measures for controlling the spread of COVID-19 infection and were at high risk of contracting and spreading the infection to their contacts [123].

Technical teams and military services were also deployed to provide additional expertise and capacity to support rural communities in the USA [124, 125]. They assisted with the establishment of testing sites, the investigation of transmission in high-risk congregate settings,

such as long-term care facilities, food processing facilities, correctional facilities, and services to persons experiencing homelessness. They also assisted with developing data collection instruments, conducting training on COVID-19 case investigation and contact tracing, and providing support to improve public health information technology systems [124, 125].

*Preparation and practices of rural health workers.* Several studies indicated that rural general practitioners and nurses across countries reported a lack of preparedness, a lack of resources including staff, limited information, guidance and training, and inadequate supplies of PPE, oxygen, water and other essential items [103–106, 113]. This contributed to fear, anxiety, psychological distress, depression, stigma, discrimination, and feeling overwhelmed. However, they also experienced collaboration, teamwork, innovation, and improvisation, recognised the importance of selfcare and the support of colleagues, built relationships with communities and improved trust [103–106, 113].

Compliance with the COVID-19 response measures and management protocols varied across contexts, with a study from Greece reporting that rural PHC workers fared well compared to urban health workers [126]. Conversely, a study from India suggested that rural providers in some locations had poor compliance with recommended COVID-19 case management practices [127].

Multiple studies described innovative training programmes to prepare rural health professionals and community workers to respond to COVID-19, largely through short courses using digital technology. Important features of the courses were rapid availability, flexibility, accessibility through online platforms, and linguistic and cultural appropriateness. Courses emphasised their evidence base, the involvement of experts, the use of simulation, the creation of social networks, peer to peer coaching and communities of practice to support active learning [122, 128–131].

Other rural innovations included expanded access to testing and care such as the erection of screening tents for testing; rapid adoption of telehealth services for COVID-19 information, screening and care; integrated web-based information systems to monitor the COVID-19 pandemic; and the creation of rapid decision making and communication structures and systems supported by digital technology in some settings [132–135]. However, rural communities in comparison to urban communities also made less use of digital COVID-19 services as illustrated in a USA study [136].

*Delivery of particular therapeutic agents in these settings.* The only identified rural studies of therapeutic agents were of the delivery of COVID-19 Convalescent Plasma and of Monoclonal Antibodies at rural facilities in the USA [137, 138]. A study from Vietnam documented the extensive use of Herbal Medicine for COVID-19 [139].

**Maintaining essential health services and systems.** The 63 studies on strategies for MEHSS focused mainly on i) Telemedicine/telehealth interventions (53; 88.3%), with relatively few on ii) maintaining or improving access to medicines, mental health care services, and mobile EMS services during the pandemic.

*Telemedicine/telehealth interventions.* The rapid increase in the use of telemedicine early in the pandemic largely compensated for the decreased in-person encounters in health care in some contexts, as evidenced in studies from the USA, Australia and Germany [140–142]. Although rural areas benefited from telemedicine, USA studies found that uptake was lower than in urban areas with many barriers and challenges including infrastructure, logistics, and internet connectivity [140, 143, 144]. Telemedicine supported a range of conditions and services including mental health, chronic diseases and care of older people, and increased access to prevention, health promotion, primary care, medical and surgical care, rehabilitation and palliative care. Settings for telehealth included patients homes, primary care services, hospitals, community based organisations, long term care facilities and prisons in rural areas.

In the USA, policy amendments reduced restrictions on telemedicine consultations and prescribing by primary practitioners, and providers rapidly improved their infrastructure and capacity to offer telehealth [145]. A local study in the USA reported that factors contributing to successful scaling up of telehealth included dedicated clinical staff, physician engagement, support of leadership, technical training opportunities, change management support, and adequate digital infrastructure and systems [146].

In addition to consultations with health care practitioners via video or telephone calls, rural telemedicine support included rapid pre-operative diagnoses and sorting of patients for surgery, [147, 148] robotic ultrasound scanning during antenatal care for pregnant women in remote areas, [149] a Tele-ICU service from specialist referral services to health facilities in rural India, [150] and remote device kiosks for screening for chronic diseases in pharmacies [151].

Studies suggested that many patients preferred audio-visual components of telehealth, versus telephone or audio only. However, some indigenous communities and older rural patients preferred face to face or telephonic options due to a lack of familiarity with digital technology, and limited access to infrastructure and equipment to support telehealth [152–155].

Studies from Australia and the USA indicated that patients and practitioners were largely satisfied with telehealth, and users reported improved physical and mental quality of life [156–159]. Higher socioeconomic status and education levels were associated with increased use of telemedicine in rural areas [155, 159–162]. Patients with more chronic conditions, needing more than one visit per annum, and travelling longer distances were more likely to use telehealth services [159, 161].

Telehealth benefits for clients included ease of scheduling and reduced wait time for appointments, quicker responses to emergencies, decreased travel time and costs, patient safety during the pandemic, compatibility with work commitments and child care arrangements, improved parental involvement and responsiveness, faster access to laboratory investigations and results, and improved continuity of care [142, 163, 164].

Benefits to health services included reduced travel time for staff, improved staff efficiency and cost savings, improved multidisciplinary teamwork, retention of services in rural communities, including retention of revenues and ongoing employment opportunities for local communities [156, 162]. Telehealth platforms also facilitated the collection of geographical and sociodemographic data on users to inform health service responses for vulnerable groups [134].

Telehealth increased opportunities for more comprehensive care including risk assessments, home based screening, behaviour change counselling, psychosocial support and pharmacy support for rural individuals and families. Two studies from Germany and the USA indicated that self-care practices using telehealth options also increased [142, 165]. Some studies reported that community and faith based organisations also adopted telehealth options including live streaming and radio to provide psycho-social support and services during COVID-19 [142, 165, 166].

A review across five LMICs found indications of readiness for mobile consulting in communities with minimal resources, but also found substantial infrastructure challenges and advised that wider system strengthening was needed to optimise the benefits of telehealth [167]. A systematic review of virtual health services in South Africa during COVID-19 found that a wide range of innovative digital health technology services had been utilised. However, these faced multiple barriers from infrastructural and technological, organization and financial, policy and regulatory barriers as well as cultural barriers, which impacted rural more than urban areas [168].

*Maintaining or improving access to medicines, mental health care services, and mobile EMS services.* Among the non-telehealth MEHSS studies, risk modelling of the functioning of the

Brazilian Mobile EMS for remote riverside and coastal areas identified critical elements, including the role of CHWs in emergencies, which enabled them to re-organise the service to better respond during the COVID-19 pandemic [169]. Medicine deliveries were facilitated by policy changes to allow 'take home' medication for longer periods for stable patients with a wider range of diagnoses; home deliveries of medication; community based 'pick up' points for refills; and the use of geospatial information to plan medication distribution in rural areas, as illustrated in studies from the USA and Uganda [170, 171]. Community based mental health care services were expanded in Mexico to respond to needs during COVID-19 by building capacity to support communities and health professionals [172].

**Vaccination.** The 18 publications on COVID-19 vaccination in rural areas reported on i) Vaccination coverage and effects, ii) Vaccine hesitancy, and iii) Strategies to improve vaccine acceptance and uptake.

*Vaccination coverage and effects.* Vaccine coverage ranged between 40% and 60% between December 2020 and May 2021, but was significantly lower in rural than urban areas in the USA [173–176]. Vaccine uptake was lowest in farming and mining dependent rural counties, in rural areas with limited health infrastructure, lower education levels, in younger age groups, and in some minority ethnic groups in rural areas [173–176]. Rural residence was also associated with lower acceptance rates of COVID-19 vaccine in health care workers in studies in Lebanon, Jordan and the USA [177–179]. In contrast vaccination coverage in people older than 50 years in Wales was high (>90%) between December 2020 and April 2021, and rural residents were more likely to be vaccinated than those in urban areas [180]. A USA study reported high vaccine effectiveness of mRNA COVID-19 vaccines in Veterans, which was similar in urban and rural areas [181].

In general, rural areas reported health care constraints that limited effective vaccination coverage, particularly shortages of appropriate health care professionals, and inadequate public health care infrastructure, including limited cold chain capacity for the transport and storage of vaccines. In LMICs these health system limitations were more extreme.

*Vaccine hesitancy.* Studies on vaccine hesitancy and communication about vaccines identified key concerns as the novelty of the vaccine, side effects or vaccine safety, the number of required doses, and reliability of the manufacturers [182–185]. Rural populations were also concerned about accessibility and availability of vaccines, particularly in LMICs [182–185]. Rural populations that obtained most of their information from the media (internet, TV, or combined sources), reportedly had more concerns about vaccines and greater vaccine hesitancy [184]. Select studies suggested that the provision of vaccine information and consultations with recommendations from health professionals and health authorities increased confidence that vaccines were safe, improved the likelihood of vaccine acceptance and increased vaccine coverage [177, 182, 185].

*Strategies to improve vaccine acceptance and uptake.* Recommended strategies for enhancing vaccine acceptance and uptake in rural areas, included information and communication strategies, the integration of COVID-19 vaccination into routine medical care, and alternative delivery sites including home delivery of vaccinations for the elderly or disabled, workplace vaccination especially on farms and factories, and at schools in rural areas [186]. 'Smart' vaccination strategies in India and South Africa defined and tested criteria to inform planning and prioritisation of vaccination in the context of vaccine scarcity and health system constraints [187, 188]. The type of vaccine used in rural areas due to transport and storage (i.e., cold chain) requirements and access to health care facilities (for repeat doses versus single dose vaccines) were important considerations [187, 188].

**Social determinants of health.** The 29 studies on SDH described i) the extent to which the COVID-19 pandemic and control efforts negatively impacted livelihoods and food

security, ii) the responses to mitigate the effects on food security and iii) the social and economic responses. A few studies also described iv) the role of digital technology in the response to food insecurity and other SDH.

*COVID-19 pandemic and control efforts' impact on livelihoods and food security*. Studies from sub-Saharan Africa and Pakistan suggested that the pandemic and control responses contributed to acute deprivation for rural, marginalised communities with loss of wages and state-imposed barriers to accessing facilities and public provisions [189, 190]. Spatial and gendered disparities in economic opportunity meant rural areas and women and children were most affected by the economic contraction [191, 192]. Select studies from South Africa and China indicated that while all areas experienced dramatic job losses at the beginning of lockdown, urban areas started to recover earlier than rural areas where unemployment persisted for longer [191, 193]. Several studies reported that COVID-19 exacerbated existing urban-rural inequities.

The pandemic affected the food security pillars of availability, accessibility, utilization, and stability of food systems. Rural communities were disproportionately impacted, as illustrated with three different studies reporting that one third to more than two thirds experienced low or very low food security in select locations [190, 194, 195]. There was variation across regions, countries and within countries depending on a number of factors, including their connectivity locally and globally, existing food systems, and local and national support and relief for agriculture and food security during the pandemic. Studies suggested that the principal Asian and Pacific farming and food systems were moderately resilient to COVID-19, reinforced by government policies that prioritized food availability and affordability [196–198].

Rural livelihoods and food security were affected primarily because of disruptions to local labour markets, farm produce markets and input supply chains (i.e., seeds and fertilisers). Increased food prices, disruptions in food logistics, food scarcity in grocery stores, the inability to work due to movement restrictions and the subsequent loss of income, and less purchasing power contributed to the decline in food security [190, 195–197, 199].

The consequences were uneven within rural communities, with women, farmers, youth, and children suffering more. Women as small farmers and in seafood value chains were more vulnerable to the market shocks associated with COVID-19 in the Pacific [196, 198]. Low income families, larger households and those that remained in quarantine also experienced more food insecurity [190].

*Responses to mitigate the effects on food security*. A common positive indicator of recovery potential was the range of strategies taken by communities and governments to support rural livelihoods [196]. Rapid prioritisation and public policy responses to support agriculture and food security strategies included the use of public funds to incentivise agricultural activity, and a range of policy incentives to increase production and home gardening [195, 196]. In select locations in Pakistan and the USA, households who received income support or aid from government, such as state-supported food assistance, experienced less food insecurity than their counterparts during the pandemic [190, 195].

Other responses included expansion of community food resources such as food banks, canteens and distribution of food parcels, home food gardening and other projects to support food provisioning [195, 199, 200]. Communities sought to maintain sufficient levels of food through food sharing, increased local food production and previous stockpiling of produce such as rice [191, 198]. This commitment of local communities and volunteering associations, as well as an increasing interest in reconnecting with local food practices, became important, as evidenced in the study from Pakistan [190].

Small-scale farmers in Southern Africa and Indonesia relied mainly on social capital by using coping strategies developed within their immediate family, farming networks, or

neighbourhoods [201]. Women farmers were better able to adapt than male farmers where women were traditionally the primary decision makers around food growing, purchase, preparation, and consumption [201]. Localized seed and input supply systems were crucial to enhancing agricultural production's resilience to future pandemics. However, small-scale farmers in some regions had little transformative capacity and agency to enable them to support local communities' food needs [201]. Studies from countries in different regions suggested that short food supply chains proved resilient and ameliorated the impact of the pandemic, with strong local connections providing support when global connections were disrupted [196–198].

Households in Pakistan adopted several coping strategies, such as rationing, dietary changes, eating less, and increased short-term food availability by borrowing food or buying food on credit [190]. Dietary changes across countries were characterised by increased consumption of vegetables and staples (maize, rice, and bread), while consumption of meat, fish, fruit, oil, and sugar decreased [197, 201]. Simple nutrition and gender messaging as part of behavioural change communication during COVID-19 improved dietary quality among women [202].

*Social and economic responses.* Several studies described social and economic support through cash transfers, emergency supplies, social prescribing, social solidarity through reciprocal practices, and technological innovations [191, 194, 203]. Different studies highlighted that the coverage of COVID-19 cash transfers appeared to benefit urban residents more than rural, with suboptimal coverage of qualifying recipients in several countries [191, 194, 203]. A study suggested that digital cash payments through the Pradhan Mantri Garib Kalyan Yojana (PMGKY) programs improved their coverage and efficiency in India [204].

Communities drew on traditional practices to strengthen community resilience during COVID-19. Osekkai, a traditional Japanese behaviour in the community, was utilized as a form of social prescribing in rural communities by creating a helping culture and social participation [205]. This was also demonstrated in the Tigani community in rural Romania where intense social cooperation, strong sense of family, community and mutual assistance helped them to fight COVID-19. It also strengthened adaptability to the societal changes and their power to keep intact their cultural identity [206]. Grassroots organizations in the Andes instituted a range of hybrid reciprocity practices as essential for provisioning during COVID-19. e.g. bartering and gift-giving [50, 207].

*The role of digital technology.* A few studies described the use of digital technology which contributed to increased levels of self-efficacy and emotional well-being in rural areas by providing access to learn new skills to generate an income or improve crops; increased accessibility to provide and seek help via social media, especially building on the tradition and practice of serving and helping others [193, 208–210]. E-agriculture and e-commerce enabled bartering in digital form such as WhatsApp to sell produce, for groups to communicate and share food, and for farmers to contact agricultural extension workers or to connect with urban markets. It was a means for social connectedness, to relate to people, talk to them, and ask for help when needed [208].

## Discussion

More studies were included from HIC's and focused on MEHSS, vaccination and clinical management, whereas studies from LMIC addressed SDH and risk communication and community engagement more. Key findings from the thematic analysis were that rural health systems were less prepared and equipped, and national COVID-19 responses were often not adequately tailored to rural areas. Promising COVID-19 responses involved local leaders and

communities, were collaborative and multisectoral, and drew on local cultures. Non-pharmaceutical interventions were applied less, support for access to water and sanitation at scale was weak, and more targeted approaches to the isolation of cases and quarantine of contacts were preferable to blanket lockdowns. Rural pharmacists, community health workers and agricultural extension workers assisted in overcoming the shortages of other health professionals. Vaccination coverage was hindered by weaker health systems in rural areas. Digital technology enabled better coordination, communication, and access to health services, yet for some was inaccessible. Determinants of health, including rural livelihoods and food security, were affected through disruptions to local labour markets, farm produce markets and input supply chains.

An important output of the study was to identify the successes and enablers, challenges and barriers and research gaps to inform future policy, practice and research on HEPR in rural and remote areas.

## Successes and enablers

Many examples of innovations or practices that facilitated COVID-19 preparation and responses in rural areas emerged. Rapid state policy responses to needs, particularly policy coordinated across sectors, expanded health care services by reducing restrictions on practitioners and services, allowed for greater use of health and digital technology, and provided support to protect rural food systems and cash transfers to vulnerable rural households.

Characteristics which contributed to successful rural COVID-19 HEPR included organisational dimensions, multidisciplinary and multi-sectoral collaborations and engagement with local leaders and communities. Responses that were contextually and culturally appropriate, and which respected and built on traditional practices and social capital, strengthened community responses and resilience.

Rural risk communication was enhanced by information from credible or trusted sources, including health professionals, health authorities, local leaders, CHW's and credible experts; and by the use of multiple channels of communication, including interpersonal communication in rural settings.

Strict isolation of cases, contact tracing and quarantining of all contacts, and shielding of vulnerable groups, appeared preferable to blanket lockdowns in rural areas. Although NPIs were effective in rural settings, rural populations tended to apply NPI preventive measures less and needed more local support for NPI's.

The re-organisation and reprioritisation of health facility services, and expansion beyond medical facilities to temporary, mobile and non-traditional sites assisted in broadening access to COVID-19 testing, care and vaccinations. Innovative online training programmes sought to prepare rural health workers, and capacity was expanded by involving and extending the roles of pharmacists, CHWs, and external technical teams. Models of multidisciplinary care emerged which were more comprehensive and sought to respond to the psychosocial needs of individuals and families.

The limited COVID-19 surveillance in rural areas was expanded by linking electronic laboratory and patient information systems where they existed and through community surveillance systems. Technology contributed extensively to rural HEPR, particularly through the use of rapid RNA tests, innovative decontamination services for PPE, and digital technology to support telehealth for COVID-19 care and the MEHSS, training, communication, coordination, social support, e-agriculture, and to support food systems and livelihoods in rural communities.

State support programs were supplemented by a wide range of community responses such as food pantries and food parcels, home and community gardens, sharing of produce, and

financial assistance directly between households, as a buffer to food insecurity and extensive loss of income in many rural settings.

## Challenges and barriers

Studies reported an urban bias in policy, financing and resourcing, both preceding and in response to the pandemic. National COVID-19 plans were often perceived as unsupportive of rural contexts and needs. Rural settings reported inadequate information, health infrastructure, human resources and supplies, which were more extreme in LMIC settings. Rural health workers frequently felt unprepared, unsupported and inadequately equipped to respond to COVID-19. Pre-existing concerns about equity of access to care in rural settings, both to primary care and for referrals for advanced hospital care, were magnified by the pandemic.

Rural areas appeared to receive less risk communication, and received it later and in forms that were less appropriate to the populations and context, reducing trust in the information. This limited the effects on behaviour change, which were moderated by local beliefs, norms and living conditions.

Barriers to telehealth existed at multiple levels (policy, organisational and individual), and included financial, regulatory, cultural and language challenges in LMIC and vulnerable rural communities in HIC. Telehealth in the COVID-19 HEPR thus inadvertently contributed to inequities in access to health care, with urban, better educated and higher socioeconomic groups benefiting most.

Insufficient planning and coordination across sectors to mitigate the impact of COVID-19 and responses on livelihoods and food security was reported in many rural settings. Civil society organisations that provided food and social assistance often had limited resources or lacked flexibility in their funding to respond to the arising needs. The impact of COVID-19 on rural livelihoods was multi-faceted, with most being disadvantaged in multiple ways, including difficulties in sustaining pre-existing livelihoods, in benefitting from policy shifts and poverty relief initiatives, and in accessing healthcare. Some rural communities, however, were relatively protected, especially where good local food distribution/sharing networks were in place and/or where agricultural support programmes were specifically targeted to small-scale farmers. The impact was uneven with vulnerable groups in rural communities experiencing greater challenges.

## Implications

This review has aimed to inform ongoing global and national processes and plans to ensure adequate preparedness and response to health emergencies. Among these processes and plans are the reviews of national capacities linked to the International Health Regulations, the National Action Plans for Health Security, One Health operations, National Disaster Risk Management Strategies, and, specific to COVID-19, national COVID-19 response and recovery plans [211–214].

The evidence on the importance of localization and decentralization of health emergency preparedness and response has been documented by independent reports, including the Independent Panel for Pandemic Preparedness and Response (IPPPR), which highlighted that "countries with the poorest results in addressing COVID-19 had uncoordinated approaches [. . .] They lacked the capacity to mobilize quickly and coordinate between national and subnational responses" [215]. Some of these lessons have already been incorporated into updated technical guidance for the implementation of the International Health Regulations (2005). The updated State Parties Annual Reporting Tool (SPAR) now incorporates indicators on gender equality, risk communication and community engagement among others [216]. The updated

Joint External Evaluation and the upcoming Benchmarks for IHR Capacities guidance, are some of several tools and guidance which incorporate elements to facilitate the operationalization of approaches that support localization of interventions and strengthening of sub-national capacities- in line with the findings of this scoping review. Moreover, WHOs *10 proposals to build a safer world together* also reflects the importance of looking at subnational capacities [217].

Important lessons from the review for HEPR policy and plans include the following:

- Pandemic preparedness, readiness and response plans need to be "rural proofed" to ensure that they adequately take into account the needs, varying contexts, and opportunities of rural areas;[218]

- Improved vertical coordination across national, regional, and local levels of government is key to better pandemic preparedness and response in rural areas, allowing for optimal adaption and tailoring for local rural contexts;

- Myth-breaking ("Covid-19 is an urban disease") is required to galvanize adequate support and attention to pandemic preparedness, readiness and response in rural areas;

- Rural community and stakeholder engagement platforms helped enable locally appropriate solutions to evolve and barriers to access services to be overcome, and should be adequately resourced;

- Many of the bottlenecks encountered with COVID-19 preparedness and response in rural areas reflect pre-existing health systems deficiencies, made worse by COVID-19, and this reinforces the call for health systems strengthening in rural and remote areas;

- Cross-governmental action for enhanced integrated and transformation rural development is required to address the social determinants of health in rural areas that were exacerbated by COVID-19 and which hindered overarching preparedness and response;

- Policymakers and planners working on integrated rural development policy across sectoral domains have a role in supporting pandemic preparedness, readiness and response.

## Limitations

This review sought to scope the literature on COVID-19 HEPR in rural and remote areas in all settings. However, we found relatively few studies on COVID-19 HEPR in LMICs and in particular rural or remote communities experiencing vulnerability in HIC settings, or studies conducted across different rural communities and/or countries. We also found very few studies on certain topics including WHO SPRP pillars of Points of entry, international travel and transport, and mass gatherings; Operational support and logistics, and supply chains, or on the financing of COVID-19 responses and financial impact on health systems, as well as on One Health. At the same time it is acknowledged that the search was completed after the first two years of the pandemic, thus additional articles of relevance released since April 2022 were not captured.

We included all study designs, but studies were largely descriptive or exploratory, with few formal evaluations or experimental studies of interventions. The findings are therefore broadly descriptive of the field and the review does not assess effectiveness of particular COVID-19 HPER's in particular communities or settings.

The challenges in interpreting rural-urban disaggregated data must be flagged: lower reported excess death or lower reported hospitalization in rural areas do not necessarily equate

with stronger health systems and less exposure to disease in these locations. They may in fact be linked to weaker reporting capacity and/or unmet need, so further reviews and consideration of these numbers must be made in the process of more thorough performance assessments.

### Further research

This scoping review identified gaps in the literature which can guide further research. More research needs to be supported on COVID-19 HEPR in rural and remote settings in LMIC's including more in-depth exploratory studies of particular rural communities or risk groups in to understand their needs and appropriate strategies. Further research is also warranted in communities living with vulnerability across HICs and LMICs, such as indigenous communities, migrant workers, refugees and isolated older adults, while also recognizing the need to evaluate operations across the full spectrum of rural and remote areas (including island and river communities, remote mountainous communities, etc) where service delivery modalities have needed to adapt to varying geography and infrastructure constraints. Studies across different communities to compare practises and interventions in different settings would also provide important lessons for the future HEPR.

Overall, more rigorous studies which are designed to evaluate the effectiveness of differentiated, decentralised approaches across diverse rural health systems are needed. Operations research underpinned by realist evaluation approaches, and participatory research methods where interventions are co-designed with communities were advocated to tailor responses to local needs and contexts.

Proposed study topics on innovative ways to improve access to COVID-19 care and MEHSS in rural areas during and post pandemic include the uptake and utilisation of telehealth, the role of a wider range of formal and informal health workers, alternative delivery sites and transportation modes. In addition, research should be undertaken on the well-being and resilience of healthcare workers, and measures needed to support them during crises.

Further research is also needed on the under-reporting of COVID-19 in rural communities, including morbidity and mortality in communities experiencing vulnerability and high-risk groups to inform the planning and monitoring of the COVID-19 pandemic and responses.

Studies on the impact of COVID-19 policy responses in rural communities should assess the longer term social and economic consequences of outbreak control measures in rural communities. These should assess alternative strategies to strengthen resilience and adaptive capabilities and explore more collaborative approaches across government levels and sectors which could reduce the negative consequences during and post pandemic.

Finally, it is important that an update of the global literature be done to cover the second biennium of COVID-19 response (2022–2024).

### Conclusions

The COVID-19 pandemic has tested the HEPR capacities of countries across the world. The evidence presented in this review on the HEPR to COVID-19 in rural and remote areas, reveal important lessons which must be incorporated into ongoing discussions about global health security. The need for decentralized and localized HEPR planning cannot be contested; evidence points towards the grave negative implications of attempting to apply policies that worked in one place to another, contextually different setting. Risk communication messages can get lost in translation and not reach those who need it most—with detrimental consequences for efforts to contain infectious disease outbreaks. Similarly, digital-based responses, which may lead to positive outcomes in some settings or with particular groups of the

population, may be ineffective in other settings where internet or mobile coverage is not as high, accessibility to digital platforms is lower, or preference for trusted communication sources lies elsewhere. The evidence also highlights the need for intersectoral action to address health needs and social determinants of health to ensure that equitable approaches to disease management and control are implemented.

More rigorous studies of HEPR in vulnerable rural and remote communities, particularly in LIC's, is needed including on the social and economic consequences of outbreaks and responses. Effective HEPR approaches should also integrate intersectoral forms of analysis and design.

Lastly, it is essential for the global community to work towards the reduction of social inequities, including through the strengthening of rural health systems, in order to be better prepared to face future infectious disease outbreaks, including of pandemic potential. Only by doing so will we be better prepared to contain and respond to the next pandemic.

## Supporting information

**S1 Checklist. PRISMA checklist.**
(DOCX)

**S1 Text. Glossary of terms.**
(DOCX)

**S1 Table. Grey literature sources.**
(DOCX)

**S2 Table. Included studies.**
(XLSX)

## Acknowledgments

Guidance was sought from WHO Health Emergencies experts, including: Jilian Sacks, April Baller, and Melinda Frost. Vittoria Lutje of Liverpool School of Tropical Medicine, United Kingdom, advised on the search strategy and conducted the electronic database searches. Innocentia Lediga, Mercia Companie and Kopano Dube, of the Department of Global Health, Stellenbosch University assisted with the data capture. Special acknowledgment goes to Pramila Shrestha, WHO, for her contribution to the commissioning of and organizational support in the early phases of this work, and to Hortense Nesseler, WHO, for reviewing the article and her project management support in its finalization.

**Disclaimer:** This article represents solely the views of the authors and in no way should be interpreted to represent the views of, or endorsement by, the World Health Organization. The World Health Organization shall in no way be responsible for the accuracy, veracity and completeness of the information provided through this article.

## Author Contributions

**Conceptualization:** Lilian Dudley, Ian Couper, Niluka Wijekoon Kannangarage, Theadora Swift Koller.

**Data curation:** Lilian Dudley, Selvan Naidoo.

**Formal analysis:** Lilian Dudley, Ian Couper, Selvan Naidoo, Clara Rodriguez Ribas.

**Funding acquisition:** Theadora Swift Koller.

**Investigation:** Lilian Dudley, Ian Couper.

**Methodology:** Lilian Dudley, Ian Couper, Niluka Wijekoon Kannangarage, Selvan Naidoo, Clara Rodriguez Ribas, Taryn Young.

**Project administration:** Ian Couper, Theadora Swift Koller.

**Resources:** Theadora Swift Koller, Taryn Young.

**Supervision:** Lilian Dudley, Ian Couper, Clara Rodriguez Ribas, Theadora Swift Koller, Taryn Young.

**Validation:** Lilian Dudley, Ian Couper, Niluka Wijekoon Kannangarage, Selvan Naidoo.

**Visualization:** Lilian Dudley, Selvan Naidoo.

**Writing – original draft:** Lilian Dudley, Ian Couper, Theadora Swift Koller.

**Writing – review & editing:** Lilian Dudley, Ian Couper, Niluka Wijekoon Kannangarage, Selvan Naidoo, Clara Rodriguez Ribas, Theadora Swift Koller, Taryn Young.

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
