## [Decision Letter · Decision Letter 0]

15 May 2023

PGPH-D-23-00631

COVID-19 preparedness and response in rural and remote areas: A scoping review.

Dear Dr. Dudley,

Thank you for submitting your manuscript to PLOS Global Public Health. After careful consideration, we feel that it has merit but does not fully meet PLOS Global Public Health’s publication criteria as it currently stands. Therefore, we invite you to submit a revised version of the manuscript that addresses the points raised during the review process.

The reviewers have highlighted several issues to take into specific consideration when revising your manuscript. Notably, please ensure to address the feedback related to the description of the methodology, and the presentation of the findings. Further details are provided below. 

We look forward to receiving your revised manuscript.

Kind regards,

Claire J Standley

Academic Editor

Journal Requirements:

1. We noticed you have some minor occurrence of overlapping text with the following previous publication(s), which needs to be addressed:

- https://doi.org/10.3390/healthcare9060621

- https://doi.org/10.1080/14733285.2021.1924359

- http://dx.doi.org/10.5888/pcd17.200254

In your revision ensure you cite all your sources (including your own works), and quote or rephrase any duplicated text outside the methods section. Further consideration is dependent on these concerns being addressed.

2. Please provide separate figure files in .tif or .eps format.

Additional Editor Comments (if provided):

Please review the reviewers' comments carefully, particularly related to additional description of the methodology (per reviewer 2) and the presentation of the findings (per reviewer 1).

Reviewers' comments:

Reviewer's Responses to Questions

**Comments to the Author**

1. Does this manuscript meet PLOS Global Public Health’s publication criteria? Is the manuscript technically sound, and do the data support the conclusions? The manuscript must describe methodologically and ethically rigorous research with conclusions that are appropriately drawn based on the data presented.

Reviewer #1: Yes

Reviewer #2: Yes

2. Has the statistical analysis been performed appropriately and rigorously?

Reviewer #1: N/A

Reviewer #2: No

3. Have the authors made all data underlying the findings in their manuscript fully available (please refer to the Data Availability Statement at the start of the manuscript PDF file)?

Reviewer #1: Yes

Reviewer #2: Yes

4. Is the manuscript presented in an intelligible fashion and written in standard English?

Reviewer #1: Yes

Reviewer #2: Yes

5. Review Comments to the Author

Reviewer #1: This is a scoping review of COVID-19 preparedness and response in rural and remote areas. It complies with the Arksey and O’Malley framework used and follows the PRISMA-ScR guidelines.

The review was done in response to a WHO request - please provide references to any WHO reports where the review was quoted or used.

Although the SPRP-based organisation of the findings is useful the review could benefit from a tabular summary of the main findings as an aid to users in delving through 586 lines of findings and 118 lines of discussion.

When listing findings from the reviewed studies, it may be useful to indicate whether there were studies with contradictory findings to the ones listed.

Line 153 mention the inclusion of a gender analysis - I couldn't find this in the results.

The included tables and graphs have several formatting problems, e.g. Table 1 category column is too narrow and the % columns too wide. Fig 3 legends are unreadable. I am not sure whether this is only a function of the review copy, but nevertheless distracted from reviewing the manuscript.

In general, this is well conducted scoping review that contains a wealth of information that would be useful to policy-makers if presented in a more accessible way.

Reviewer #2: Thank you for giving me the time to review your manuscript. This manuscript is exciting and scientifically meaningful for considering COVID-19 preparedness and response in rural and remote areas. Regarding the contents, the following revision should be considered.

Abstract

Structured summary:

The Abstract was provided comprehensively. The authors should add the research method in the abstract.

Introduction

-The authors should include more focused research questions and purpose for this scoping review. What kinds of HEPR should be focused in this review?

-The definition of rural and remote contexts should be described in the background.

-The background has too few paragraphs. The authors should focus on theory building, the problems, and the research question paragraphs. The first paragraph should focus on general information regarding nursing research and interests in international contexts. Moreover, the second and third paragraphs should introduce the gap between nursing practice and research and the research question as the theoretical and conceptual framework.

-The introduction should include the international contexts and research questions of this study.

-This study should describe why the review process used the suggested search engines to review comprehensively.

Methods

-What kinds of languages can be used for searching literature? English only? or used some translated materials?

-Line 154, “research questions” what is this research question? The authors should describe it in the background clearly.

-The explanation of deductive thematic analysis should be thicker. The present explanation has a lot of vague point regarding the qualitative analysis.

Result

-Figures are unclear in quality. The authors should revise the quality of the figures.

-Tables and figures contain some abbreviations. They should be spelled out.

-How did the researchers perform the thematic analysis? which framework of thematic analysis was used?

Discussion

-The discussion should start by summarizing the result of the study and suggesting discussion points.

-The discussion should describe more regarding this review's outstanding points. Moreover, the discussion part needs paragraph writing.

6. PLOS authors have the option to publish the peer review history of their article (what does this mean?). If published, this will include your full peer review and any attached files.

**Do you want your identity to be public for this peer review?** For information about this choice, including consent withdrawal, please see our Privacy Policy.

Reviewer #1: **Yes: **Kobus Herbst

Reviewer #2: No

---

## [Author Response · Author response to Decision Letter 0]

17 Jul 2023

Editor We noticed you have some minor occurrence of overlapping text with the following previous publication(s), which needs to be addressed:

- https://doi.org/10.3390/healthcare9060621

- https://doi.org/10.1080/14733285.2021.1924359

- http://dx.doi.org/10.5888/pcd17.200254

Response: Two of these articles are included as studies in the review and are referenced (Shahzad 2021, Jamieson 2021). The third is a commentary by McElroy 2021 which we did not include. We have rechecked the Turnitin reports to locate the particular texts, have included additional references to sources and rephrased parts of the text to address any duplications. 

Editor: Please provide separate figure files in .tif or .eps format 

Response: Figures 1 and 2 are now provided in tif format as per the journal guidelines

Editor Please review the reviewers' comments carefully, particularly related to additional description of the methodology (per reviewer 2) and the presentation of the findings (per reviewer 1)

Response: Please see responses below. We have expanded on the methods, and have modified Table 2 to include a summary of key results. 

Reviewer 1 Please provide references to any WHO reports where the review was quoted or used.

Response: As the review manuscript is not yet in the public domain, WHO reports are not yet quoting it. That said, presentations on the work have been done in WHO-convened platforms like the COVID-19 mortality TAG working group 5 on inequalities, amongst others.

Reviewer 1 The review could benefit from a tabular summary of the main findings as an aid 

Response: We have now replaced table 2 with an extended table which includes a summary of key results for each subtheme and theme.

Reviewer 1 It may be useful to indicate whether there were studies with contradictory findings to the ones listed.

Response: As far as possible we have reported contradictory findings in the results.

Reviewer 1 Line 153 mention the inclusion of a gender analysis - I couldn't find this in the results. 

Response: Most of the included studies did not provide a gender analysis, and where it was provided we have reported on it. However, it is a limitation and we will remove the stated intent as it was not adequately achieved.

Reviewer 1 The included tables and graphs have several formatting problems, e.g. Table 1 category column is too narrow and the % columns too wide. Fig 3 legends are unreadable We have reformatted the Table 1 and increased the font sizes on figure 3 to address the reviewers concerns.

Reviewer 2 Add the research method in the abstract.

Response: We have included a statement that we used the Arksey and O’Malley approach for scoping reviews. 

Reviewer 2 Introduction: include more focused research questions and purpose for this scoping review. What kinds of HEPR should be focused in this review? 

Response: We have clarified the main aim and purpose of the review in the introduction. As stated, the scoping review covered all HEPR guided by the WHO SPRP for COVID-19 in rural and remote settings. As per definition of a scoping review, the intent is ‘to map the evidence and identify main concepts, theories, sources and knowledge gaps'. The findings can be used to define further more focused research questions for systematic reviews.

Reviewer 2 The definition of rural and remote contexts should be described in the background 

Response: As there are multiple conflicting definitions of rural and remote in different settings, we chose to accept all definitions used by authors. This was previously explained in a sentence under the methods section, which we have now moved to the introduction

Reviewer 2 The background has too few paragraphs. The authors should focus on theory building, the problems, and the research question paragraphs. The first paragraph should focus on general information regarding nursing research and interests in international contexts. Moreover, the second and third paragraphs should introduce the gap between nursing practice and research and the research question as the theoretical and conceptual framework.

Response: We have expanded on the problems, aims and purpose of this study. However we are not clear why the reviewer is referring to nursing research as this was not the topic of this scoping review. We are therefore unclear whether these comments were intended for our manuscript or perhaps mistakenly included.

Reviewer 2 The introduction should include the international contexts and research questions of this study We refer to the responses above. This study provides a global perspective on HEPR and searched for studies in rural areas from all HIC & LMIC settings as stated in the introduction, and described further in the methods section. 

Reviewer 2 Describe why the review process used the suggested search engines to review comprehensively 

Response: At the protocol stage we had input from the information specialist on both search terms and databases to search. A sentence has been included to indicate the reasons broadly for the selection of the particular search engines.(line 136) 

Reviewer 2 What kinds of languages can be used for searching literature? English only? or used some translated materials? Response: The information specialist’s guidance included on covering all languages, for example non-English records identified were in French, Italian, Chinese, and Korean. All had English abstracts which were used during screening, and we translated the full texts of any articles which were selected for inclusion. We have therefore indicated in the methods that we sought studies in all UN languages (line 124). Under the study selection section we also indicated that we translated abstracts and full texts into English for screening and data extraction.(line 157)

Reviewer 2 Line 154, “research questions” what is this research question? The authors should describe it in the background clearly.

Response: We have clarified the purpose, aims and objectives of the study in the methods, and have now referred to these. (line 182) As per purpose of the scoping review, the initial research question was broad, and has been captured under aims and objectives. 

Reviewer 2 The explanation of deductive thematic analysis should be thicker. The present explanation has a lot of vague point regarding the qualitative analysis.

Response: The deductive analysis was guided by the WHO SPRP categories, and additional themes provided in consultation with WHO. (line 185) We have clarified this further in the text. We have also expanded on the approach to the inductive analysis within the main themes, for which we used a grounded theory approach by coding the contents of the included studies within each theme. 

Reviewer 2 Results: Figures are unclear in quality. The authors should revise the quality of the figures. 

Response: As per response to the editor and reviewer 1, we have updated the figures to the tiff format and increased the font size.

Reviewer 2 Tables and figures contain some abbreviations. They should be spelled out.

Response: Abbreviations have been written in full in the figures and tables.

Reviewer 2 How did the researchers perform the thematic analysis? which framework of thematic analysis was used? 

Response: As per explanation above, we used both a deductive framework (SPRP categories +) and grounded theory to identify the main subthemes within each of the main themes. 

Reviewer 2 Discussion: The discussion should start by summarizing the result of the study and suggesting discussion points.

Response: We have included a summary of the results, and indicated that the further discussion focuses on challenges, barriers and future research. 

Reviewer 2 The discussion should describe more regarding this review's outstanding points. Moreover, the discussion part needs paragraph writing.

Response: There is a dedicated section on “limitations” in the discussion that highlights some of the outstanding points, followed by forward-looking considerations on those in the next section on “further research”. These sections contain paragraphs.

---

## [Decision Letter · Decision Letter 1]

13 Sep 2023

PGPH-D-23-00631R1

COVID-19 preparedness and response in rural and remote areas: A scoping review.

Dear Dr. Dudley,

Thank you for submitting your manuscript to PLOS Global Public Health. After careful consideration, we feel that it has merit but does not fully meet PLOS Global Public Health’s publication criteria as it currently stands. Therefore, we invite you to submit a revised version of the manuscript that addresses the points raised during the review process.

Please review the suggestions provided by Reviewer 3 (see attachment), which may improve the flow and clarity of the Introduction in particular. If the editorial management system provides an opportunity to describe the role of the funder in the research, please be sure to do so there (particularly since it appears that some co-authors may be affiliated with the funding entity). Otherwise, please be sure to clarify the role of the funder and the process of participating in the commissioned research in the manuscript text.

We look forward to receiving your revised manuscript.

Kind regards,

Claire J Standley

Academic Editor

Journal Requirements:

1. We noticed you have some minor occurrence of overlapping text with the following previous publication(s), which needs to be addressed:

- https://doi.org/10.3390/healthcare9060621

- https://doi.org/10.1080/14733285.2021.1924359

- http://dx.doi.org/10.5888/pcd17.200254

In your revision ensure you cite all your sources (including your own works), and quote or rephrase any duplicated text outside the methods section. Further consideration is dependent on these concerns being addressed.

Additional Editor Comments (if provided):

Please see minor suggested edits below from Reviewer 3, particularly with respect to clarifying aspects of the Introduction. The manuscript will not require re-review after these revisions are addressed.

Reviewers' comments:

Reviewer's Responses to Questions

**Comments to the Author**

1. If the authors have adequately addressed your comments raised in a previous round of review and you feel that this manuscript is now acceptable for publication, you may indicate that here to bypass the “Comments to the Author” section, enter your conflict of interest statement in the “Confidential to Editor” section, and submit your "Accept" recommendation.

Reviewer #1: All comments have been addressed

Reviewer #3: All comments have been addressed

2. Does this manuscript meet PLOS Global Public Health’s publication criteria? Is the manuscript technically sound, and do the data support the conclusions? The manuscript must describe methodologically and ethically rigorous research with conclusions that are appropriately drawn based on the data presented.

Reviewer #1: Yes

Reviewer #3: Yes

3. Has the statistical analysis been performed appropriately and rigorously?

Reviewer #1: Yes

Reviewer #3: N/A

4. Have the authors made all data underlying the findings in their manuscript fully available (please refer to the Data Availability Statement at the start of the manuscript PDF file)?

Reviewer #1: Yes

Reviewer #3: Yes

5. Is the manuscript presented in an intelligible fashion and written in standard English?

Reviewer #1: Yes

Reviewer #3: Yes

6. Review Comments to the Author

Reviewer #1: My original comments have been addressed. I have no further comments on this version of the manuscript.

Reviewer #3: (No Response)

7. PLOS authors have the option to publish the peer review history of their article (what does this mean?). If published, this will include your full peer review and any attached files.

**Do you want your identity to be public for this peer review?** For information about this choice, including consent withdrawal, please see our Privacy Policy.

Reviewer #1: **Yes: **Kobus Herbst

Reviewer #3: No

---

## [Editor Report · Decision Letter 2]

24 Oct 2023

COVID-19 preparedness and response in rural and remote areas: A scoping review.

PGPH-D-23-00631R2

Dear Prof Dudley,

We are pleased to inform you that your manuscript 'COVID-19 preparedness and response in rural and remote areas: A scoping review.' has been provisionally accepted for publication in PLOS Global Public Health.

Best regards,

Claire J Standley

Academic Editor

Thank you for the clear and detailed responses to the editorial and reviewer comments.